# Multi-Drug Resistance Bacterial Infections in Critically Ill Patients Admitted with COVID-19

**DOI:** 10.3390/microorganisms9081773

**Published:** 2021-08-20

**Authors:** Daniela Pasero, Andrea Pasquale Cossu, Pierpaolo Terragni

**Affiliations:** 1Department of Medical Surgical and Experimental Science, University of Sassari, 07100 Sassari, Italy; pterragni@uniss.it; 2Department of Emergency, Anaesthesia and Intensive Care Unit, AOU Sassari, 07100 Sassari, Italy; andreapasquale.cossu@aousassari.it

**Keywords:** hospital acquired infection, multi-drug resistance, COVID-19, SARS-CoV-2, secondary bacterial infections, critically ill patients

## Abstract

Introduction. It is known that bacterial infections represent a common complication during viral respiratory tract infections such as influenza, with a concomitant increase in morbidity and mortality. Nevertheless, the prevalence of bacterial co-infections and secondary infections in critically ill patients affected by coronavirus disease 2019 (COVID-19) is not well understood yet. We performed a review of the literature currently available to examine the incidence of bacterial secondary infections acquired during hospital stay and the risk factors associated with multidrug resistance. Most of the studies, mainly retrospective and single-centered, highlighted that the incidence of co-infections is low, affecting about 3.5% of hospitalized patients, while the majority are hospital acquired infections, developed later, generally 10–15 days after ICU admission. The prolonged ICU hospitalization and the extensive use of broad-spectrum antimicrobial drugs during the COVID-19 outbreak might have contributed to the selection of pathogens with different profiles of resistance. Consequently, the reported incidence of MDR bacterial infections in critically ill COVID-19 patients is high, ranging between 32% to 50%. MDR infections are linked to a higher length of stay in ICU but not to a higher risk of death. The only risk factor independently associated with MDR secondary infections reported was invasive mechanical ventilation (OR 1.062; 95% CI 1.012–1.114), but also steroid therapy and prolonged length of ICU stay may play a pivotal role. The empiric antimicrobial therapy for a ventilated patient with suspected or proven bacterial co-infection at ICU admission should be prescribed judiciously and managed according to a stewardship program in order to interrupt or adjust it on the basis of culture results.

## 1. Introduction

Whether critically ill patients affected by coronavirus disease 2019 (COVID-19) are at higher risk of bacterial infections has already been a matter under discussion. The main concern is related to the time of infection development, whether early at admission with SARS-CoV-2 infection or later during the intensive care unit (ICU) stay, which is indeed per se a risk factor of developing hospital acquired infections. Bacterial infections are commonly reported at admission in cases of viral respiratory tract infection such as influenza and its incidence is highly variable: around 20–30% of patients are diagnosed with severe influenza [1,2,3]. The coexistence of viral and bacterial infections at admission is usually associated with greater severity of illness and increased risk of death [4]. Therefore, both the severity of the respiratory disease in patients affected by COVID-19 at ICU admission and the difficulty of ruling out a bacterial co-infection on presentation lead to a wide prescription of broad-spectrum antimicrobial drugs. Moreover, the extensive introduction of empiric antimicrobial therapy was endorsed by the Surviving Sepsis Campaign guidelines, which suggested the use of empiric antimicrobial treatment over no treatment at ICU admission in ventilated patients [5]. Several retrospective analyses reported a quite low incidence of bacterial infections at admission compared to previous epidemic viral infections, such as two different meta-analyses that reported an incidence of co-infections between 3.5% (95%CI 0.4–6.7%) and 14% (95% CI 5–26%) in the ICU context; therefore, the prescription of antimicrobial therapy should be reduced and carefully evaluated only for specific cases [6,7]. However, many patients affected by COVID-19 developed hospital acquired infections during their ICU stay, further increasing the severity of organ dysfunction, especially in cases complicated by septic shock that showed doubling mortality rates [8]. COVID-19 can be associated with a significant and sustained lymphopenia compromising the immune system; in fact, a significant decrease in lymphocyte and increase in neutrophil count has been described, especially in the most severe cases, which presented with an inflammatory storm. The overall incidence of bacterial infections in patients admitted with a diagnosis of COVID-19 has been estimated at 6.9% (95%CI 4.3–9.5%), while the incidence increased to 8.1% (95% CI 2.3–13.8%) in critically ill patients [6,7], who are ventilator-dependent and need invasive monitoring and pharmacological support by central lines [9]. However, a small proportion of bacterial infections were described as co-infection, defined as those reported at the time of COVID-19 diagnosis, and existing at patient’s admission (3.5%, 95% CI 0.4–6.7%). Most of these infections were caused by community-acquired pathogens such as *Streptococcus pneumoniae* and *Haemophilus influenzae*, as it had already been described during the previous viral epidemic infections [4,6,7]. Most bacterial infections described in the COVID-19 population emerged during the ICU and hospital stay, were defined as secondary bacterial infections and their incidence was estimated around 14.3% (95% CI 9.6–18.9%) [6], as shown in Figure 1.

### Pathophysiology and Main Clinical Pattern of Superinfections

It is not well understood which is the main underlying mechanism of SARS-CoV-2 infection in determining the pathophysiology that drives to over-infections. Regarding bacterial infections, one of the hypotheses is that the viral infection causes a direct damage of the lower airway epithelial cells associated with an impaired mucociliary clearance that facilitates bacterial binding to cell surfaces [10]. A similar mechanism was already described during influenza viral infections. The consequent inhibition of the epithelial cells’ mechanisms of repair and regeneration caused by bacterial infection determines a further aggravation of the damage. Nevertheless, many other factors such as airway cell receptors up-regulation, decreased immune response or abnormal release of inflammatory mediators may also play a crucial role during these early stages [11,12]. Furthermore, the administration of the protonic pump inhibitor (PPI) during an ICU stay has been associated with the rise of secondary infections and the pathophysiological mechanism might be sustained by the suppression of the gastric acid production, which might increase gastric microbiota and small intestine bacterial overgrowth [13]. Therefore, the mechanism associated with lung microaspiration might lead to bacterial colonization of the lower respiratory tract and an increase of pneumonia [14].

It is also reported that SARS-CoV-2 directly infects human gut enterocytes with resulting disruption of the gut barrier. This might increase the intestinal permeability and facilitate bacterial translocation, one of the most important risk factors for the development of blood stream infections (BSI) [15]. The hypothesis of a direct damage of the enteric barrier caused by the viral infection was previously described in SARS-CoV-1 infection; however, further studies are needed to better understand the pathophysiology mechanism that causes gut barrier disruption [16].

Moreover, there is an increasing number of studies reporting a probable or possible correlation between fungal and SARS-CoV-2 infections. Patients with acute respiratory distress syndrome have an increased risk of developing invasive aspergillosis, even in the absence of underlying immunocompromised conditions [17,18]. Galactomannan from bronchoalveolar lavage fluid is the most reliable and useful diagnostic tools for invasive pulmonary aspergillosis, [19,20,21,22,23]. However, bronchoscopy use should be limited in COVID-19 patients to avoid the aerosol-dissemination intrinsic to the procedure that has a consequent increased risk of viral transmission to the staff members and other patients [24]. Previous lung disease, prolonged mechanical ventilation and immunosuppressive therapy are risk factors for aspergillosis development [25,26]. COVID-19 patients may also be co-infected by other respiratory viruses, especially influenza, respiratory syncytial virus and adenovirus, which are the most common viral pathogens reported. However, the pathophysiology mechanism has not been well understood and data available from recent published studies are not enough to understand whether a viral co-infection affects the prognosis of COVID-19 patients [7,27,28]. Recently, Alosaimi et al. in a study including 48 COVID-19 patients, reported that Influenza AH1N1 was the most common detected virus among the co-infecting pathogens, found in 64% of patients. Clinical relevance of influenza AH1N1 co-infection is not easy to determine because the clinical characteristics are very similar to SARS-CoV-2 infection [29].

The simultaneous spread of two viruses clearly emphasizes the importance of screening for other clinically important co-circulating respiratory pathogens. Influenza AH1N1 co-infection was associated with increased severity of clinical manifestations and ICU admission.

In the present review, we tried to understand more clearly the role of secondary bacterial infections in critically ill patients affected by COVID-19 and describe the possible risk factors and mechanisms that contributed to increase the incidence of multidrug resistant (MDR) pathogens, during SARS-CoV-2 pandemic.

## 2. Methods

We performed a literature search for the present study on the lines of search for a narrative review, but with some structural features of systematic review methodology as reported in Figure 2 [30,31]. The review was conducted to select all indexed papers that reported data on secondary bacterial infections caused by multi-drug-resistant germs in COVID-19 critically ill patients. We searched through PubMed/MEDLINE between January 2020 and April 2021 and only for papers published in the English language. We privileged retrospective, single or multi-center studies, prospective observational studies, and randomized clinical trials, which reported data on MDR secondary infections in critically ill adult patients admitted with COVID-19. We excluded studies reporting only co-infections, i.e., bacterial infections reported at hospital or ICU admission, fungal or other opportunistic infections; in addition to this, also excluded were review articles, systematic review and meta-analysis, abstracts, case reports and case series articles. Two independent authors (D.P. and A.P.C.) selected titles based on terms that took into consideration inclusion and exclusion criteria: “COVID-19”, “secondary infection”, “bacterial infection”, “multi-drug resistance”, and “critically-ill patient”. Each selected paper was analyzed, and the type of study and sample size were described, together with few demographic variables and only those reporting information on MDRO were summarized in a table. The main microorganisms responsible for antimicrobial resistance were reported: *Enterococcus faecium*, *Staphylococcus aureus*, *Acinetobacter baumannii*, carbapenem resistant *Enterobacterales*. Additional information on potential risk factors for secondary infections and multi-drug resistance, such as steroids administration, and broad spectrum antibiotic empirical therapy at admission, were extracted.

## 3. Results

Among 2803 papers published in journals indexed on PubMed/MEDLINE and reporting in the title the key words COVID-19 and secondary infection, we selected 246 articles that reported any data on secondary bacterial infections in critically ill patients admitted to intensive care units with a diagnosis of COVID-19. All selected articles were analyzed independently by D.P. and A.P.C. to include only full-text articles on human studies, that were retrospective or prospective clinical trials, and single-center or multicentric which reported any data on MDR secondary infections in critically ill COVID-19 patients; therefore, 12 final studies were selected and summarized in Table 1. We excluded from the final selection any other data coming from case reports or case series or other reviews, and systematic reviews or meta-analysis, to avoid at least a partial duplication of the reported data due to different selection criteria. All selected studies were retrospective and single centered, except two Italian articles, which were two retrospective multicenter studies involving hub hospitals with a bigger sample size compared to most single-centered studies. Three of them were case-control studies where the authors selected the control group from a matched pool of septic patients without COVID-19. From each study we reported the sample size, which varied from site to site, while the median age of the studied population was similar and was between fifty and seventy years of age. The majority of the patients received an empiric antibiotic therapy at intensive care unit admission, between 69% and 97%; types of molecules and spectrum were rarely specified. Most of the selected studies reported information on steroids administration, with a wide variation among them, from 4.7% to 77%, any other immunosuppressive therapy was rarely reported. The incidence of MDR infections was between 30% and 50% for the secondary bacterial infections and most of them were caused by GRAM negative bacteria, predominantly by *Carbapenem resistant enterobacterales (CPE)* and *Pseudomonas aeruginosa* (Table 1).

## 4. Discussion

Although several studies reported data on superinfections in COVID-19 patients, some of them did not distinguish between early and late infections, which is not only a semantic dilemma, but it is a real need related to antimicrobial prescription to reduce the uncontrolled empiric administration in favor of a more judicious evaluation. More recently, most of the retrospective analysis and two meta-analyses highlighted that co-infections are rare, while the majority are acquired infections that developed later during a hospital and ICU stay [6,7,8]. Several studies reported that critically ill patients are at higher risk of developing secondary infections, mainly due to invasive devices such as the central venous catheter and endotracheal tube, and the prevalent sites of healthcare associated infections involve the respiratory tract and the blood stream, 10–15 days after admission [8,38,41,42]. However, the real incidence of healthcare associated infections, especially in critically ill COVID-19 patients, was probably underestimated because of the competing risk with death, which was higher and it might have occurred early during the ICU stay before the development of a secondary infection, due to the severity of the disease associated with a condition of shock [8,43].

### 4.1. Multidrug Resistance Organisms: Development and Spread during the COVID-19 Outbreak

The extensive use of broad-spectrum antimicrobial drugs during the COVID-19 outbreak might have contributed to the selection of pathogens with different profiles of resistance. All microorganisms such as *Staphylococcus aureus*, *Enterococcus spp*., *Enterobacteriaceae*, *Pseudomonas aeruginosa* and *Acinetobacter* spp. are often responsible for healthcare associated infections and prone to develop multidrug resistance. According to definitions of the European Center for Disease Prevention and Control (ECDC) and of the Center for Disease Control and Prevention (CDC), MDR infection is described as an acquired infection non-susceptible to at least one agent in three or more antimicrobial categories [44]. An incidence of MDR infections in critically ill COVID-19 patients ranging between 32% to 50% was reported by several papers [8,32,35,36,38,45]. A group of Italian authors tried to explain the possible mechanisms that led to an increase of multi-drug resistant strains among common bacteria, responsible for secondary infections in the COVID ICU, in a country where the spread of antibiotic resistant bacteria was very high and largely attributed to carbapenem resistant *Klebsiella pneumoniae* (CR-Kp) [46]. They retrospectively analyzed all microbiological samples of rectal swabs collected for routine surveillance before and during the pandemic to compare CPE colonization incidence. The authors observed an increase of CPE colonization up to 50% in patients admitted to their ICU with COVID-19. Interestingly, they observed that patients who underwent prone-supine position had higher incidence of CPE transmission compared to those who did not need that type of treatment, which involved several health care workers (HCW) each time the maneuver had to be performed. Moreover, they observed that the use of personal protection equipment (PPE) to protect HCW from SARS-CoV-2 transmission and the need to employ a larger number of HCW, who were not trained for intensive care medicine, might have reduced the awareness of hand hygiene, contact precautions and PPE disinfection when moving from one patient to another; the latter were regular practices described in an antimicrobial stewardship program (ASP) implemented in their ICU before the outbreak and they were in accordance with the ESCMID guidelines for the management of infection control measures to reduce transmission of multidrug resistant Gram-negative bacteria in hospitalized patients [47]. Furthermore, work overload of ICU staff, ICU overcrowding, reinforcement of less experimented staff and PPE shortage were described as potential additional factors leading to a decrease in adherence to infection control measurements, allowing the spread of MDROs [48,49,50]. Additionally, other authors reported a PPE shortage such as gowns with the need to use the same for more than one patient [36].

### 4.2. MDR Secondary Infections and Risk Factors

Most of the published studies on COVID-19 patients with complications of secondary infections were retrospective and reported single-centered data on a small number of patients, but the incidence and the time of development of the secondary infections were almost homogeneous (Table 1). Karulli A. et al. reported 16 out of 32 patients developing an MDR infection with the prevalent isolates of CPE, and the highest mortality rate of 72%. Although the sample size was small, some considerations might be made about the high number of patients on empirical antibiotic therapy at admission (78%) and the high percentage of patients that were on steroids (53%) [32]. Similarly, Patel A. et al. reported a spread of MDR secondary infections caused by *Gram negative bacteria*; the whole sample of patients described had an MDR infection and the majority of infections reported were caused by CPE (46%). Moreover, nearly the whole studied population was on an empiric antibiotic therapy upon admission (97%) compared to the study by Karulli, and 35% of them received steroids [33]. Baiou A. et al. reported 33% of MDR secondary infections with around 10% of the isolated microorganisms as carbapenem-resistant *Klebsiella pneumoniae* and *Pseudomonas aeruginosa*. In the latter, differently from the two previous studies, patients were younger and few of them received steroids (4.7%). The only risk factor independently associated with MDR secondary infections was the invasive mechanical ventilation (OR 1.062; 95% CI 1.012–1.114) [34]. A reduction of antibiotic prescriptions has been described between January 2020 and April 2020 (OR 0.28, 95% CI 0.08–0.98) and a lower prescription was also observed in studies evaluating children (38.5%, 95% CI 26.3%–52.3%) when compared to studies including only adult population (83.4%, 95% C.I. 76.6–88.5%), while a higher antibiotic administration was described with the increasing of median or mean patient age, per 10 years increase (OR 1.45, 95 CI 1.18–1.77) and in patients requiring invasive mechanical ventilation (OR 1.33, 95% CI 1.15–1.54), with a 10% increase [51]. Cultrera R. et al. reported the microbiological characteristics of a high number of isolates in secondary infections on a small sample of patients: the incidence of MDR-acquired infections was similar to other retrospective studies. They found a higher frequency of methicillin-resistant *Staphylococcus Aureus* (MRSA), vancomycin-resistant *Enterococcus faecium* and carbapenem resistant *Acinetobacter baumannii* [35]. Bogossian E.G. et al. compared the incidence of multi-drug resistant organisms (MDRO) between patients admitted with COVID-19 and a control group of patients with subarachnoid hemorrhage (SAH), complicated by sepsis, admitted in a period before the outbreak. The incidence and characteristics of MDR secondary infections reported were similar to most of the data coming from the published articles on COVID-19 and did not show any significant difference in the multivariable competing risk analysis (sHR 1.71 (CI95% 0.93–3.21)) between the two groups; therefore, they concluded that COVID-19 was not an independent risk factor associated with MDR infections acquisition [36]. Nasir N. et al. reported that patients who were severe to critically ill at the time of admission with COVID-19 had an over fourfold probability to develop secondary infections (OR 4.42; 95% CI 1.63–11.9). In fact, the majority of those patients needed more invasive devices such as an endotracheal tube and central venous catheter; undergoing more frequent to invasive mechanical ventilation and, as a consequence, they were more frequently admitted to the ICU compared to the controls. Moreover, the authors showed that all patients developing secondary infections received the empiric antibiotic therapy at admission and an elevated percentage (92%) were on steroids treatment, which was an independent risk factor strongly associated with secondary infections (OR 4.6; 95% CI 1.24–17.05) [37]. Giacobbe D.R. et al. and Bonazzetti C. et al. both reported data on the incidence of blood stream infections (BSI) in critically ill COVID-19 patients [38,39]. The patients described in both papers were similar in the number of cases and age, but Bonazzetti C. et al. reported a higher number of MDR infections, caused especially by *Enterococcus spp.*, with a higher rate of mortality (49% vs. 26%). Moreover, they observed that patients who developed BSI during their ICU stay had a higher sequential organ failure assessment (SOFA) score (9.5, IQR 8–12 vs. 8, IQR 5–10), but they did not report whether their patients were on an empirical antimicrobial therapy at ICU admission and if they received steroids [39]. Instead, Giacobbe D.R. et al. reported that 96% of their patient sample were on an empiric antimicrobial treatment at ICU admission and 31% received steroids: they found that the anti-inflammatory treatment was an independent risk factor for BSI development (caused-specific hazard ratio [csHR] 3.95; 95% CI 1.20–13.03 for methylprednisolone and csHR 10.69; 95% CI 2.71–42.17, for methylprednisolone plus tocilizumab) [38]. Interestingly, Luxemburger H. et al. reported that the use of proton pump inhibitors (PPI) (OR 2.37; 95% CI 0.08–5.22) and gastroesophageal reflux disease (OR 6.40; 95% CI 1.50–35.51) were independent predictive factors of developing secondary infections in patients affected by COVID-19. These results underlined the role of microaspiration in the pathogenesis of secondary bacterial infections of the lower respiratory tract [14]. Furthermore, a short term and current use of PPI during admission for COVID-19 increased the worst-outcome risk (OR 1.90; 95% CI 1.46–2.77 and OR 1.79; 95% CI 1.30–3.10) due to an increased probability of ICU admission, invasive mechanical ventilation, and death [52]. A recent meta-analysis confirmed such findings on a bigger population coming from seven papers (OR 1.46, 95% CI 1.34–1.60) [53]. Ripa et al. reported data on both BSI and low respiratory tract infections (pLRTI) among 86/731 (11.8%) admitted to the ICU [40]. Critically ill patients more frequently developed a secondary infection. During the follow-up of 1,318 patient-days (PDFUs), 40/731 patients (5.5%) had 51 secondary infections during ICU stay with an incidence rate of 38.7 (28.8–50.9) per 1000 PDFUs, which was considerably higher when compared to patients outside the ICU, with an incidence rate of 4.0 (2.9–5.5) per 1000 PDFUs. Among factors associated with an increased risk for secondary infections, there was ICU admission within the first 48 h from hospital admission (HR 2.51, 95% CI 1.04–6.05), together with decreasing PaO_2_/FiO_2_ ratio < 100 (HR 3.67; 95% CI 1.24–10.90). The results confirmed the median time for the first secondary BSI at 13 days from hospital admission with a prevalence of Gram-positive among the isolates (*Staphylococcus epidermidis* and *Enterococcus faecium*), while at 16 days for the first pLRTI, mainly caused by Gram-negative MDRO (carbapenem-resistant *Acinetobacter baumannii*, CP-Kp, CPE and carbapenem-resistant *Pseudomonas aeruginosa*). The authors observed an increase of antimicrobial resistance compared with an historical cohort of patients before the SARS-CoV-2 outbreak, with a higher incidence rate of Gram negative MDRO among BSIs, mainly due to *Acinetobacter baumannii*, while among Gram-positive isolates there was an increase of vancomycin-resistant *Enterococcus faecium*.

Only two multicenter retrospective studies reported the incidence and risk factors for secondary infections in critically ill patients [8,41]. The former, published by Grasselli G. et al. evaluated patients with severe COVID-19 admitted to eight Italian hub hospitals [8]. They reported data on a sample of 774 patients, in which 35% developed MDR infections, among 359 patients (46%) that developed hospital acquired infections (HAI). The most frequent secondary infections were ventilator associated pneumonia (VAP, N = 389 [50%]), with 26 (95% CI 23.6–28.8) per 1000 patient-days of invasive mechanical ventilation, and BSI (N = 183 [34%]), with 11.7 (95% CI 10.1–13.5) per 1000 ICU patient-days. They observed an incidence of MDR infections like most of the single center, and in the majority of the cases the isolated microorganisms were *Enterobacterales* with resistance to carbapenems and *MRSA*, especially in VAP. The number of patients that were on empirical antibiotic therapy was similar to the other single-center studies (69%), but nearly half of these patients received broad spectrum drugs. Notably, at the multivariable analysis they showed that treatment with broad spectrum antibiotics was independently associated with HAI (HR 0.61; 95% CI, 0.44–0.84), which paradoxically described a protective role of antimicrobial treatment against the development of secondary infections; however, they reported that the policy of all participating ICUs was to interrupt empiric antimicrobial therapy early if culture results were negative for bacterial infections at admission. In addition, the authors observed that patients complicated with septic shock resulted in a doubled mortality rate (52%). The other multicenter study, published by Giacobbe D.R. et al. involved nine Italian hub hospitals that admitted 586 patients with severe COVID-19 during the first wave of the outbreak between February and May 2020. Among them, 171 (29%) patients developed an infection with diagnosis of VAP. The incidence rate of VAP was slightly lower when compared to the study by Grasselli et al. (18 vs. 26 events per 1000 ventilator days). However, the incidence of MDR infections was similar and very close to all the other studies that reported this data. The most frequently isolated germs were *Pseudomonas aeruginosa* (35%), *Staphylococcus aureus* (23%) and *Klebsiella pneumoniae* (19%). Among the MDR microorganisms, the most frequently isolated species were methicillin-resistant *S. aureus* (10%) and carbapenem-resistant Gram-negative bacteria (32%). They expressed the outcome of their population at 30 days as a 30-day case-fatality and it was quite high (78/171, 46%), but it was evaluated only on patients who developed a secondary infection. Other studies reported the mortality rate over the whole COVID-19 population; therefore, no conclusion could be drawn comparing the outcome of the results among the selected studies. Interestingly, the authors found that septic shock and ARDS at VAP onset were strongly associated with an increase of 30-day case-fatality (OR 3.30; 95% CI 1.43–7.61 and OR 13.21; 95% CI 3.05–57.26 respectively). This association was confirmed when they analyzed the subgroup of patients who developed VAP with the BALF culture positive. The incidence rate of VAP reported in patients for both studies was surprisingly high, especially when compared to that related to non-COVID-19 critically ill patients (one out of 19 episodes per 1000 ventilator days) [54,55]. The increase of VAP incidence rate in COVID-19 critically ill patients might be explained by multiple triggering mechanisms: (i) one of the main hypothesis might be that the viral infection causes a direct damage of the lower airway epithelial cells associated with impaired mucociliary clearance, which allows bacterial binding to cell surfaces [10]; (ii) a second hypothesis might be an up-regulation of the airway cell receptors [10]; (iii) a third reason might be a decreased immune response or the abnormal release of inflammatory mediators during the early stages [11,12]. Furthermore, some authors speculated on the hypothesis that there might have been an overestimation during the COVID-19 period because in the majority of cases the diagnosis was clinical and performing a BALF was rare, especially at the beginning of the outbreak. This was due to the high risk of viral transmission from the patient to the operator despite the presence of PPE, as in several hospitals in Northern Italy where a shortage was reported during the first wave.

Lastly, Li J. et al. reported 102 (6.8%) who developed secondary bacterial infections among COVID-19 patients. Similarly to most of the selected studies, they observed that critically ill patients were more prone to develop secondary bacterial infections because they received invasive mechanical ventilation and had a higher rate of central catheter placement. Among patients who developed secondary infections, 69 cases (4.6%) were caused by MDRO, with carbapenem resistant *Acinetobacter baumannii* and *Klebsiella pneumoniae* as the most frequently isolated infection found mainly in critically ill patients. Almost half of the patients who acquired secondary infections died and the critically ill patients showed an increased mortality rate 45/69 (65%) compared to less severe patients, 5/33 (15.2%) [42].

### 4.3. Limits of the Study

The present review had some limitations. First, it is not a systematic review, therefore we did not provide a statistical analysis that gives more strength to selected studies. Moreover, we did not provide a detailed incidence and description of the mechanisms of resistance, because most of the studies did not report them, due to the retrospective collection of the data. Furthermore, we might underestimate the real incidence of MDRO, because of the scarcity of microbiological samples, especially BALF for LRTI, the execution of which might increase the risk of viral transmission among the operators.

## 5. Conclusions

Critically ill patients admitted with COVID-19 are at high risk of developing healthcare associated secondary infections, especially of bacterial origin, with a variable rate of resistance profiles. The main sites of healthcare infections are the lower respiratory tract, associated with invasive mechanical ventilation, and the blood stream, mainly catheter-related infections. The incidence rate of lower respiratory tract infections was interestingly higher when compared to those related to non-COVID-19 critically ill patients, mainly due to a pathophysiological mechanism mediated by the viral infection. The strongest risk factors associated with secondary infections were mainly a critically ill condition and the need of invasive treatment together with the extensive use of corticosteroids. Main risk factors for ventilator associated pneumonia were the use of PPI and the gastroesophageal reflux disease, and in BSI a strong association with the corticosteroids use alone in association with tocilizumab, as reported by a single center study [38]. The only risk factor associated with MDR secondary infections reported was the invasive mechanical ventilation (OR 1.062; 95% CI 1.012–1.114). Moreover, the worst outcome in critically ill patients admitted with COVID-19 was associated with septic shock and ARDS at the time of VAP onset. Finally, broad-spectrum empiric antimicrobial treatment, commenced at ICU admission, might have acted as a protective factor against HAI. Therefore, we could conclude that an empiric antimicrobial treatment at admission, in critically ill COVID-19 patients, might be considered when correctly managed within a stewardship program, because it could play a protective role in case of co-infections, especially when a patient has other risk factors for MDR, such as hospital admission within the previous six months.

## Figures and Tables

**Figure 1 microorganisms-09-01773-f001:**
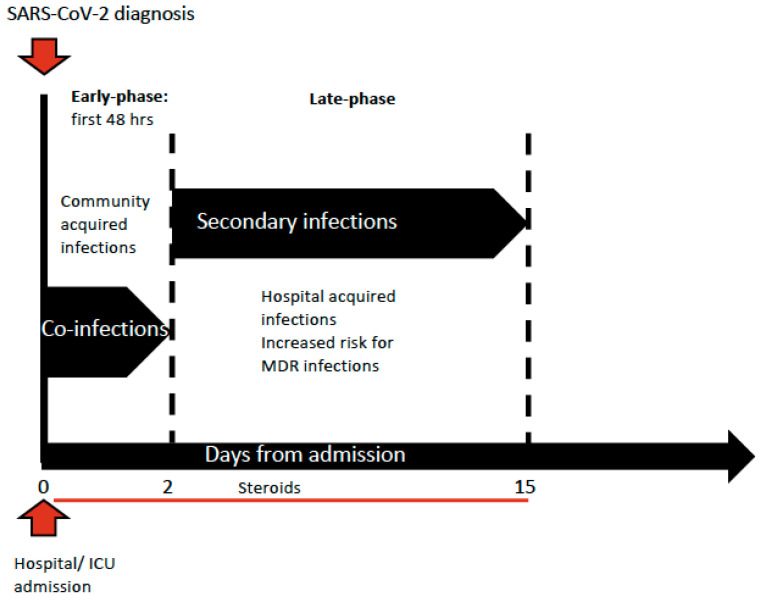
Timing of infections’ development during ICU stay.

**Figure 2 microorganisms-09-01773-f002:**
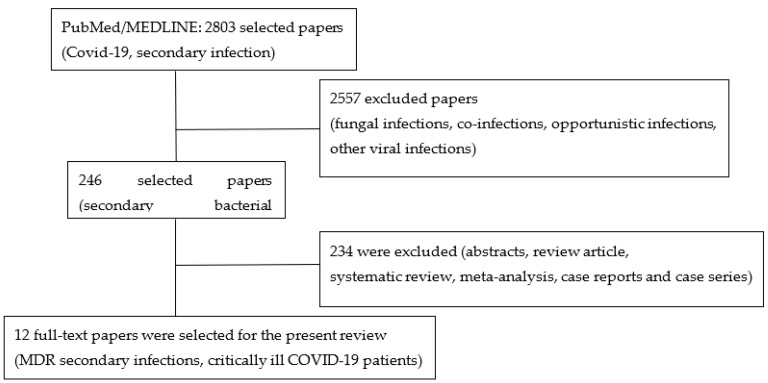
Flow diagram of the literature selection process for the present article.

**Table 1 microorganisms-09-01773-t001:** Summary of the main studies reported data on MDR secondary infections in COVID-19 critically ill patients.

Study	Type of Study	Country of the Study	Sample Size	Age Median (IQR)	*Empiric Antibiotic Therapy at Admission N (%)*	*Steroid N (%)*	*Any MDR Bacterial Infection N (%)*	*CPE Infections N (%)*	*Acinetobacter Baumannii Infections N (%)*	*Pseudomonas Aeruginosas Infections N (%)*	*MRSA N(%)*	*Enterococcus Species N (%)*	*ICU-LOS, Days Median (IQR)*	*ICU */In-hospital Mortality N (%)*
**Karulli A. et al.** [32]	Retrosp, single-center	Italy	32	68 (55–75)	25 (78.1)	17 (53.1)	16 (50)	5 (32)	3 (19)	3 (19)	1 (6)	2 (13)	10.5 (5.7–17)	23 (71.8) *
**Patel A. et al.** [33]	Retrosp, single-center	Maryland, USA	71	N-A.	69 (97)	25 (35)	71 (100)	33 (46)	27 (38)	27 (38)	N.A.	N.A.	11 (7–20)	27 (38) *
**Baiou A. et al.** [34]	Retrosp, single-center	Qatar	234	49 (40–60)	NA	11 (4.7)	78 (33)	2 (2.5)	0	6 (7.7)	N.A.	N.A.	31 (20–48)	12 (15) *
**Cultrera R. et al.** [35]	Retrosp, single-center, case-control study	Italy	28	N.A.	N.A.	N.A.	9 (32)	4 (44)	17 (61)	3 (33)	5 (56)	24 (86)	N.A.	N.A.
**Bogossian E. G. et al.** [36]	Retrosp, single-center, case-control study	Belgium	72	61 (14)	56 (78)	7 (10)	24 (33)	3 (9)	0	4 (13)	0	3 (10)	11 (3–28)	22 (31) *
**Nasir N. et al.** [37]	Retrosp, single-center, case-control study	Pakistan	100	60 (52–70)	82 (82)	77 (77)	28 (56)	0	16 (32)	5 (10)	5 (10)	2 (4)	9 (6–14)	30 (30)
**Giacobbe D.R. et al.** [38]	Retrosp, single-center	Italy	78	66 (57–70)	75 (96)	24 (31)	N.A.	0	0	0	6 (13)	8 (18)	N.A.	20 (26) *
**Bonazzetti C. et al.** [39]	Retrosp, single-center	Italy	89	61 (53–69)	N.A.	N.A.	32 (36)	10 (8.5)	1 (0.8)	1 (0.8)	9 (7.6)	53 (45)	12 (8–18)	44 (49.4) *
**Ripa M. et al.** [40]	Retrosp, single-center	Italy	731	64 (55–76)	N.A.	483 (66.1)	21 (2.8)	2 (10)	9 (43)	5 (24)	1 (4.8)	4 (19)	N.A.	194 (26.5)
**Grasselli G. et al.** [8]	Retrosp, multicentric	Italy	774	62 (54–68)	534 (69)	207 (27)	272 (35)	72 (26)	19 (2.4)	34 (4.4)	83 (31)	29 (4)	14 (8–26)	234 (30) *
**Giacobbe D.R. et al.** [41]	Retrosp, multicentric	Italy	171	64 (57–71)	162 (95)	108 (63)	60 (35)	25 (15)	N.A.	27 (16)	8 (5)	N.A.	N.A.	78 (46) *
**Li J. et al.** [42]	Retrosp, single-center	Wuhan, China	102	66 (30–93)	99 (97.1)	NA	69(4.6)	32 (31)	50 (49)	7 (4.4)	3 (1.9)	6 (3.8)	N.A	50 (49)

MDR, multi-drug resistant; CPE, Carbapenems Resistant Enterobacterales; MRSA, Methicillin Resistant Staphylococcus Aureus; ICU-LOS, Intensive Care Unit Length of Stay; * ICU mortality.

## Data Availability

No new data were created or analyzed in this study. Data sharing is not applicable to this article.

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
