# Peer review of "Multi-Drug Resistance Bacterial Infections in Critically Ill Patients Admitted with COVID-19"

_microorganisms, 2021, doi:10.3390/microorganisms9081773_

Round 1
Reviewer 1 Report
The review form pasero and colleagues examines (as a systenmtic review almost) MDR bacterial infections in critically ill patients admitted with COVID-19. It is a well written paper, though it could really beenfit from being reduced in length, especially the discussion, as the overall conclusions are not unexpected. HAI appears to be the critical issue still.
Minor comments: Introduction could be shortened, especially where the authors are writing about aspergillosis (just needs a mention not text detailed lines 96-116).
Table 1; the country that the study was done in should be inlcuded as a column in the Table.
Discussion - far too long, as the main conclusion is that ctritically-ill patients developed more frequently a secondary infection - not unexpected. Need to reduce the discussion by 50%.
No examination of the limitations of the study; these need to be added into your reduced discussion.
Author Response
Dear Reviewer,
please see the attachment below.
Best Regards
DP

Reviewer 2 Report
Pasero et al. present a review article describing current data on MDR bacterial infection in COVID-19 patients with critical illness. In general, the manuscript contains a lot of useful data and interesting details, but it is not well structured, which greatly reduces its readability. I have several comments to be addressed before the publication of this manuscript.
Major comments:
- I think that it would be better to give some definition, or a set of definitions, for ‘critically ill’ patients (e.g., in Introduction). This term is very general and its meanings may vary (e.g., patients with a high risk of death, with high score according to some scale, patients on mechanical ventilation, patients in ICU etc.)
- Discussion section is hard to follow. I suggest adding more sub-sections to make it more reader-friendly.
Minor comments:
Line 86 – “Therefore, the mechanism associated” – it is not clear how the colonization of respiratory tract is associated with low intestinal bacteria (previous sentence).
Line 114 – “where it more frequently is reported” – it would be better to rephrase like “where it is reported more frequently”
Line 119 – should be “pathogens”, not “pathogen”
Line 172 – should better be “in journals”, not “on journals”
Author Response
Dear Reviewer,
Please see the attachment below.
Best Regards
DP
